# ReViT: A Hybrid Approach for BCLC Staging of Hepatocellular Carcinoma Using 3D CT with Multiple Instance Learning

Shun-Cheng Chang[1], Hsin-Pei Yu[1], Yi-Hsien Hsieh[1], Pochuang Wang[2], Weichung Wang[3],
Tung-Hung Su[4], Jia-Horng Kao[4] and Che Lin[1,5,6,7,*]

Email: {r11942137, r11942197, d11942002}@ntu.edu.tw, d08922016@csie.ntu.edu.tw, wwang@math.ntu.edu.tw,
tunghungsu@ntuh.gov.tw, {kaojh, chelin}@ntu.edu.tw

*Abstract*—Deep learning has revolutionized medical imaging, offering advanced methods for accurate diagnosis and treatment planning. The BCLC staging system is crucial for staging Hepatocellular Carcinoma (HCC), a high-mortality cancer. An automated BCLC staging system could significantly enhance diagnosis and treatment planning efficiency. However, we found that BCLC staging, which is directly related to the size and number of liver tumors, aligns well with the principles of the Multiple Instance Learning (MIL) framework. To effectively achieve this, we proposed a new preprocessing technique called Masked Cropping and Padding(MCP), which addresses the variability in liver volumes and ensures consistent input sizes. This technique preserves the structural integrity of the liver, facilitating more effective learning. Furthermore, we introduced ReViT, a novel hybrid model that integrates the local feature extraction capabilities of Convolutional Neural Networks (CNNs) with the global context modeling of Vision Transformers (ViTs). ReViT leverages the strengths of both architectures within the MIL framework, enabling a robust and accurate approach for BCLC staging. We will further explore the trade-off between performance and interpretability by employing TopK Pooling strategies, as our model focuses on the most informative instances within each bag.

*Index Terms*—Medical Imaging, Hepatocellular Carcinoma, BCLC staging, Convolutional Neural Network, Vision Transformer, Multiple Instance Learning

## I. INTRODUCTION

HEPATOCELLULAR CARCINOMA (HCC) is a critical cancer type in medical research due to its high mortality rate and complex treatment requirements. As the most common type of primary liver cancer, HCC accounts for a significant number of cancer-related deaths worldwide [1]. The high mortality rate is largely attributed to late-stage diagnosis and the limited effectiveness of current treatment options [2]. Early and accurate staging of HCC is crucial for determining the most appropriate therapeutic strategy and improving patient outcomes.

In recent years, the application of deep learning in medical imaging for liver cancer has increased significantly. Traditionally, convolutional neural networks (CNNs) [3] have been employed for feature extraction, excelling in capturing useful information from local neighborhoods. However, the advent of Vision Transformers (ViTs) [4] introduced a new paradigm, utilizing multi-head self-attention (MSA) blocks to learn global dependencies within the image data.

The Barcelona Clinic Liver Cancer (BCLC) staging system is pivotal in managing HCC [5]. It classifies patients based on the number of tumors and the maximum diameter of the largest tumor in the liver, aiding clinicians in planning appropriate treatment strategies. Radiologists typically review each slice of a patient's CT scan to determine the number of tumors and their maximum diameter, a process that is both time-consuming and labor-intensive. An automated BCLC staging system would significantly reduce the radiologists' workload and enhance efficiency.

Despite its potential benefits, research on automated BCLC staging systems is scarce, resulting in a limited availability of public datasets for this specific task. BCLC staging requires comprehensive three-dimensional structural information for accurate patient-level diagnosis. However, the scarcity of 3D CT images with patient-level labels poses a significant challenge in developing reliable models. This highlights the need for more robust studies to advance the field. Additionally, 3D abdominal CT images are large, and downscaling these volumes can lead to significant information loss, which is detrimental for BCLC staging. Segmentation models are often used to extract the liver ROI, but resizing the ROI can distort the images due to varying liver sizes among patients, negatively impacting model training.

Multiple Instance Learning (MIL) [6] offers a promising solution for such challenges by treating an image as a collection of instances and using pooling techniques to aggregate instance-level information into a comprehensive bag representation. To better align with the MIL framework and preserve critical anatomical details, we propose a new data preprocessing method, Masked Cropping and Padding (MCP), specifically tailored for the BCLC staging task. This method maintains the integrity of the liver features while ensuring

[1] Graduate Institute of Communication Engineering, National Taiwan University (NTU), Taipei 10617, Taiwan
[2] Department of Computer Science and Information Engineering, NTU, Taipei 10617, Taiwan
[3] Institute of Applied Mathematical Sciences, NTU, Taipei 10617, Taiwan.
[4] Division of Gastroenterology and Hepatology, Department of Internal Medicine, National Taiwan University Hospital, Taipei 10048, Taiwan
[5] Department of Electrical Engineering, NTU, Taipei 10617, Taiwan
[6] Center for Advanced Computing and Imaging in Biomedicine, NTU, Taipei 10617, Taiwan
[7] Smart Medicine and Health Informatics Program, NTU, Taipei 10617, Taiwan
* Corresponding author

This project is supported by the National Science and Technology Council, the Ministry of Health and Welfare, and the Ministry of Education, Taiwan. The grant numbers are MOST 110-2221-E-002-112-MY3, MOHW112-TDU-B-221-124003, and 113L900701, respectively.

consistent input dimensions, thereby enhancing the model's ability to focus on the most informative instances.

Furthermore, inspired by the principles of MIL, we introduce ReViT, a hybrid model designed for BCLC staging. ReViT combines the local feature extraction capabilities of CNNs with the global context modeling of ViTs within the MIL framework. This integration provides a robust and accurate approach to address the challenges of BCLC staging.

## II. RELATED WORKS

### A. Deep Learning in Liver Cancer Classification

Deep learning has significantly advanced liver cancer research, particularly through the classification of liver cancer using abdominal CT images. Most studies focus on 2D imaging to identify and categorize focal liver lesions. A common approach involves segmenting the liver region of interest (ROI), masking, and resizing it to ensure consistent input dimensions [7]–[10].

However, resizing might distort images due to varying liver sizes, negatively impacting model training. Tailored pre-processing methods are used to address this, especially for tasks like the BCLC staging, which requires patient-level predictions. Fu et al. [11], using the same dataset as ours, also addresses a similar task but differs by ignoring the 3D structure of the liver. It independently assigns patient-level labels to each 2D slice, potentially missing crucial 3D contextual information. In this work, We maintain the 3D context to capture the complete anatomical structure and provide a comprehensive understanding of liver cancer.

### B. Advances in CNN and ViT for Medical Imaging

In the realm of medical imaging, both Convolutional Neural Networks (CNNs) [7], [12], [13] and Vision Transformers (ViTs) [14] have made significant contributions. CNNs excel at capturing local spatial features within images and are recognized for their efficacy in both training and deployment, rendering them optimal for real-time applications. Conversely, CNNs may encounter difficulties in grasping long-range dependencies. In contrast, ViTs demonstrate proficiency in learning long-range dependencies and capturing global context. Nonetheless, ViTs can incur significant computational costs during training and deployment, making them less conducive to real-time applications.

Recent research [15]–[19] indicates that hybrid architectures combining CNN and ViT layers effectively capitalize on the strengths of both models, addressing the CNN's deficiency in capturing global contextual information and the ViT's tendency to miss local feature details. Swin-Transformer [20] introduces a hierarchical architecture and a shifted window mechanism, effectively capturing both long-range dependencies and local interactions. MaxViT [21] integrates multi-axis attention mechanisms, block and grid-based spatial attention, combined with local and global convolutions, to effectively capture both fine-grained details and broader contextual information in images. Both have shown remarkable performance across various visual tasks. However, our proposed ReViT diverges from these approaches by not merely enhancing the

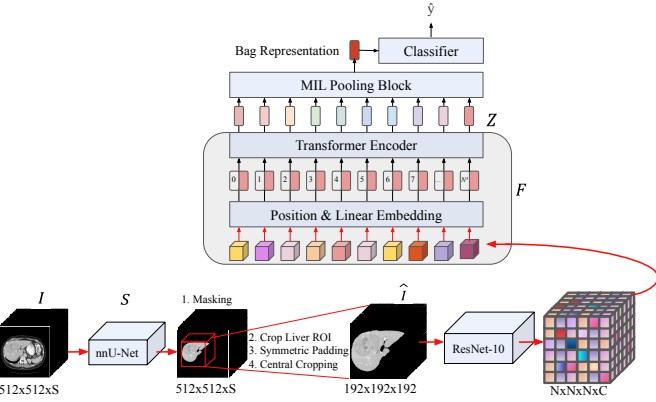

Fig. 1. The workflow of the proposed BCLC staging system using ReViT.

CNN's strengths within the transformer blocks. Instead, we chose to integrate ResNet10 to ensure that the cube instances encapsulate the detailed structures of the liver and tumors. This approach leverages the self-attention and positional encoding mechanisms of ViTs to implicitly learn critical features, such as tumor size and count, specifically tailored to meet the demands of the BCLC staging task.

### C. Multiple Instance Learning

Multiple Instance Learning (MIL) is a paradigm well-suited for scenarios where precise annotations are scarce. In the MIL framework, an image is considered a 'bag' containing multiple 'instances' (patches or cubes). Many medical imaging studies have adopted MIL, primarily focusing on 2D high-resolution histopathology images [22]–[24]. However, recent efforts have expanded its application to other areas. For instance, Araújo et al. [25] explored various vision models within the MIL framework to handle 2D dermoscopy and mammography images, achieving more clinically relevant visualizations for skin cancer and breast cancer, respectively. Han et al. [26] proposed AD3D-MIL, extending MIL to 3D CT imaging to enhance diagnostic capabilities by effectively leveraging the 3D structural information inherent in these scans. This approach employs CNN architectures combined with attention-based MIL pooling to identify key instances within the data. AD3D-MIL has been successfully applied for the automated multi-class classification of COVID-19 from chest CT images, distinguishing between common pneumonia, no pneumonia, and COVID-19, thereby demonstrating its utility in complex diagnostic scenarios. However, their method treats each instance as independent, overlooking the inherent structural relationships. In contrast, our proposed ReViT addresses this limitation by first using a ResNet backbone to extract informative cube instance representations. These representations are then fed into the ViT, which excels in capturing long-range spatial interactions and leveraging positional encoding to maintain the structural context. This integration ensures that the model not only identifies key instances but also preserves the spatial relationships crucial for accurate BCLC staging.

## III. METHODS

### A. Overall Framework

Our method, as illustrated in Figure 1, demonstrates the

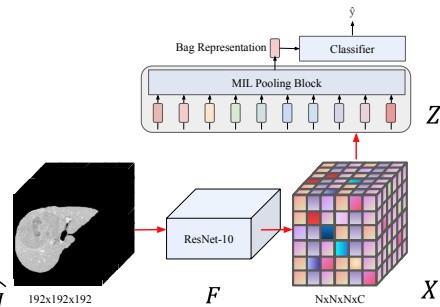

Fig. 2. The workflow of the CNN-based model within the Multiple Instance Learning (MIL) framework.

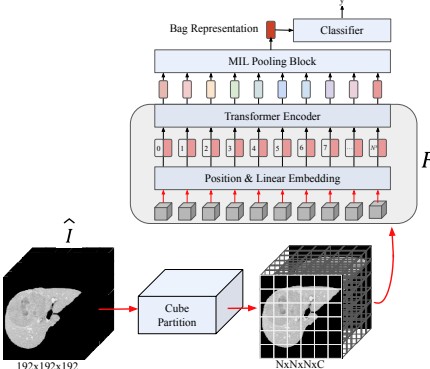

Fig. 3. The workflow of the Vision Transformer (ViT) within the Multiple Instance Learning (MIL) framework.

workflow of an automated BCLC staging system. The input 3D CT volume $I$ is processed by the Segmentation Model $S$ to obtain the liver region ROI. After appropriate data preprocessing, the liver ROI is fed into the Cube Encoder Block $F$ to extract liver features. Finally, we employ a MIL Pooling Block at the final stage to explore the impact of different MIL pooling strategies on performance and interpretability. In the following subsections, we will further discuss each component in detail.

*1) Segmentation Model:* Given a 3D CT volume $I$ with dimensions $(H \times W \times D)$, where $H$, $W$, and $D$ represent the height, width, and depth of the volume respectively. The segmentation model $S$ processes the 3D CT volume $I$ as input, resulting in a segmented output liver mask $\hat{I} = S(I)$. For this task, we adopted the nnU-Net [27], which has demonstrated promising performance across various medical image segmentation tasks.

To simulate real-world conditions where clinicians do not annotate the liver mask for each patient, we trained the liver segmentation model on the publicly available MSD [28] dataset. We then performed inference on two additional datasets, **TCIA-TACE-SEG** and **OP**. The resulting liver mask $\hat{I}$ is used as the ROI for subsequent analysis. This approach avoids noise from other organs in the entire 3D CT volume, thereby forcing the model to learn the inherent features specifically from the liver for the downstream classification task.

While BCLC staging is directly related to liver tumors, we utilize the entire liver ROI for several reasons. Firstly, the accuracy of liver area segmentation is typically higher than that of tumor area segmentation. By focusing on the

liver ROI, we aim to mitigate the propagation of errors from the segmentation model to the downstream BCLC staging task. Additionally, BCLC staging considers whether there is vascular invasion within the liver. Using the liver ROI enables the model to capture such crucial information.

*2) Cube Encoder Block:* Within the MIL framework, the Cube Encoder Block plays a crucial role. It processes the liver ROI by dividing it into $N \times N \times N$ cube instances. For each cube instance, the Cube Encoder Block generates a representation vector, capturing the essential features required for subsequent analysis. Mathematically, this can be represented as:

$$X = F(\hat{I}),$$

where $X \in \mathbb{R}^{N \times N \times N \times C}$. Here, $C$ denotes the number of channels in the representation vector, which encapsulates the learned features of each cube instance.

Next, we will introduce three distinct feature extraction strategies implemented within the Cube Encoder Block: CNN Encoder, ViT Encoder, and our proposed ReViT, and explain how we integrate these strategies into the MIL framework.

***CNN Encoder*** employs a series of 3D convolutional layers to perform feature extraction, resulting in a final feature map $X$. Each voxel in this feature map aggregates information from its corresponding receptive field in the original image, effectively capturing the characteristics of each cube instance within the CT volume. The typical CNN-based approach in classification tasks involves applying Global Average Pooling to this final feature map, producing a single representation vector for the entire image. However, within the MIL framework, other general pooling methods can be used to aggregate the instance-level features into a comprehensive bag representation, as illustrated in Figure 2.

***ViT Encoder***, in contrast, treats each cube instance as a token, utilizing self-attention mechanisms to capture global context. This process is facilitated by the linear embedding function, which converts each cube instance into a token that the ViT can process. This mechanism allows the model to understand long-range dependencies and interactions within the liver ROI, effectively leveraging global contextual information for more accurate analysis. In the standard ViT architecture, a learnable class (CLS) token is prepended to the sequence of tokens. This CLS token is connected to the classification head to obtain the final image classification, effectively harnessing information from all cubes. However, to better align with the MIL framework, we remove the CLS token and utilize each individual token directly for the MIL Pooling Block, as illustrated in Figure 3.

***ReViT*** is a hybrid approach (see Figure 1) that combines the strengths of both methods: the *CNN Encoder* serves as the Cube Embedding Block for the *ViT Encoder*, first extracting features of each cube instance, which are then fed into the *ViT Encoder* as tokens. This integration leverages the detailed local feature extraction capability of CNNs and the global contextual understanding provided by ViTs. The ResNet backbone allows the model to learn informative cube instance representations early, effectively capturing the features of both tumors and

the liver. Subsequently, the ViT utilizes positional encoding along with a series of multi-head self-attention mechanisms to capture the spatial relationships among the cube instances. This approach ensures that the model not only maintains fine-grained local details but also understands the broader contextual information, aligning with the requirements of the BCLC staging task. Incorporating these mechanisms enhances the interactions among cube instances, providing a more comprehensive and nuanced feature representation and ultimately improving diagnostic accuracy and robustness.

However, both the Global Average Pooling in CNNs and the CLS token in ViTs share a similar concept: they aggregate information across the entire input space to form a bag representation. This approach assumes that all input parts are equally relevant to the task. Therefore, in the next section, we will introduce the design of the MIL Pooling Block and explore other MIL pooling strategies beyond Global Average Pooling, which provide different perspectives on aggregating instance-level information within the input.

*3) MIL Pooling Block:* After we use the Cube Encoder Block to conduct a series of feature extractions, we obtain a bag of cube instance representations, capturing the essential features of localized regions within the liver ROI. Mathematically, this set of representations can be denoted as $\mathbf{X} \in \mathbb{R}^{N \times N \times N \times C}$, where $N \times N \times N$ represents the number of cube instances and $C$ is the number of features per instance. Flattening $\mathbf{X}$ yields $\mathbf{Z}$, where $\mathbf{Z} = \{\mathbf{z}_1, \mathbf{z}_2, \ldots, \mathbf{z}_{N^3}\}$ and each $\mathbf{z}_i$ is a row vector in $\mathbb{R}^C$, representing the features of the $i$-th cube instance. This transformation facilitates the application of various pooling strategies within the MIL framework.

Next, we implement two MIL strategies, including Global Average Pooling and TopK Pooling, to explore their impact on performance and interpretability.

Global Average Pooling aggregates the features by computing the mean of all instance representations, providing a balanced view of the entire liver ROI. Mathematically, if the set of instance features is $\{z_1, z_2, \ldots, z_{N^3}\}$, then the average pooling can be represented as:

$$\mathbf{Z}_{\text{avg}} = \frac{1}{N^3} \sum_{i=1}^{N^3} z_i.$$

This method assumes that all regions within the liver contribute equally to the final diagnosis, which can be beneficial for capturing a holistic understanding but may dilute the impact of the most critical features.

In contrast, TopK Pooling targets the most diagnostically informative instances by selecting the top-k instances with the highest activation values from the feature map and averaging them. Specifically, if we denote the top-k highest-activating instance features as $\{z_{(1)}, z_{(2)}, \ldots, z_{(k)}\}$, where $z_{(i)}$ represents the $i$-th highest activation, then the TopK Pooling can be represented as:

$$\mathbf{Z}_{\text{top-k}} = \frac{1}{k} \sum_{i=1}^{k} z_{(i)}.$$

This approach aims to highlight the most significant regions within the liver, forcing the model to base the final bag representation on only a few of the most critical instances.

After applying the selected pooling strategy, the pooled representation $\mathbf{Z}_{\text{pool},n}$ is passed through a softmax classifier to obtain the final predicted probabilities for each class:

$$\hat{\mathbf{y_n}} = \text{softmax}(\mathbf{W}\mathbf{Z}_{\text{pool},n} + \mathbf{b}),$$

where $\hat{\mathbf{y_n}}$ is the predicted probability vector for the $n$-th bag. This prediction will be used in conjunction with the corresponding ground truth label to compute the loss function during model training.

In our experiments, we will further analyze these strategies to provide a comprehensive understanding of their effects on the BCLC staging task.

*B. Loss Function*

In this study, we employ the Cross-Entropy Loss function for our multi-class classification task, which involves three classes corresponding to the BCLC staging system. This loss function effectively measures the difference between the predicted probabilities and the ground truth labels.

Let $\hat{y}$ represent the predicted probability distribution over the three BCLC stages, and let $y$ be the ground truth label. The Cross-Entropy Loss $L$ is defined as:

$$L = -\frac{1}{N} \sum_{n=1}^{N} \sum_{i=1}^{3} y_{n,i} \log(\hat{y}_{n,i}),$$

where $y_i$ is the ground truth probability for class $i$ (1 for the true class, 0 for the others), and $\hat{y}_i$ is the predicted probability for class $i$. Minimizing this loss function during training enhances the model's accuracy in predicting the correct BCLC stage.

IV. EXPERIMENT SETTING

*A. Datasets*

The scarcity of datasets specifically curated for BCLC staging poses a significant challenge. To address this, we utilized two distinct datasets: (1) ***TCIA-TACE-Seg*** and (2) ***OP***, initially compiled for different purposes, resulting in varying distributions. We combined these datasets to enhance model training and facilitate comprehensive performance comparison. Below, we introduce these datasets and summarize their distributions in Table I.

*1) TCIA-TACE-Seg:* We collected 105 multiphasic abdominal CT images from the open dataset TCIA [29], specifically from patients who underwent transarterial chemoembolization (TACE) for HCC at the University of Texas MD Anderson Cancer Center between 2002 and 2012. This open dataset was initially designed to leverage radiomics techniques to predict whether patients would be TACE-susceptible or refractory. However, since the clinical data includes BCLC staging labels as determined by clinical experts, we used this dataset for the automated BCLC staging models. To ensure consistency with the ***OP*** dataset, we only used the portal venous (PV) phase images from this dataset.

*2) OP:* This in-house dataset consists of 147 abdominal CT images annotated with BCLC staging by clinical physicians. The dataset, spanning from 2015 to 2021, focuses on clinical experts' evaluation of HCC patients deemed eligible for surgical operations. The CT images were acquired during the PV phase, which is typically the most informative phase for radiologists to identify the precise location of HCC. This phase allows for the accurate measurement of tumor size and count, which are critical for determining the BCLC stage.

## B. Data Preprocessing

In the datasets mentioned above, all 3D CT volumes have dimensions of $512 \times 512 \times S$, where $S$ represents the number of slices varying across volumes. For instance, in the ***TCIA-TACE-Seg*** dataset, $S$ ranges from 25 to 98, with spacing between 2.5 to 5.0 mm. In the ***OP*** dataset, $S$ ranges from 5 to 77, with spacing between 2.0 to 5.0 mm. To address potential noise from variable spacing during training, we resampled each CT volume to a consistent voxel size of $1.5 \times 1.5 \times 1 \,\text{mm}^3$ and filtered out some outlier images to maintain data quality and consistency. Additionally, following Tang et al. [20], we clipped the intensity values of the original CT volume to the range of $[-21, 189]$ Hounsfield Units (HU) and normalized them to the range of [0, 1] using the mean and standard deviation from the training set.

Some studies [7], [10], [30] utilized a method called *Resized Masking*, where the liver mask predicted by the segmentation model was used to directly mask the original CT volume. Given the original CT volume dimensions of $512 \times 512 \times 512$, the size was too large for practical processing. To standardize the input size for the model, these masked volumes were resized to fixed dimensions of $128 \times 128 \times 128$. However, this resizing led to significant information loss and distortion of the liver structures, adversely affecting the model's training performance and accuracy.

Due to the complexity and variability in BCLC staging, the MIL framework is particularly well-suited for this task. The MIL approach enables the model to partition the liver region into numerous cube instances, focusing on instances containing critical diagnostic information such as tumor presence and characteristics. To optimize the input data for MIL, we developed a new preprocessing technique termed *MCP*.

We crop a bounding box around the liver based on the liver mask. To address the variability in liver volumes among patients, we apply symmetric padding to ensure any cropped liver image smaller than $192 \times 192 \times 192$ is padded to these standardized dimensions. This padding technique ensures the liver remains centrally located within the padded volume. We then implement central cropping to trim any excess regions from the padded images, resulting in a uniform size for all liver images. This preprocessing approach preserves the liver's structural integrity and ensures consistent input sizes across all samples, enhancing the learning process. By maintaining this standardized size, the method better fits the MIL framework, allowing the model to focus on the most informative instances within each bag and improving the accuracy and robustness of BCLC staging.

| | TCIA | | | OP | | |
|---|---|---|---|---|---|---|
| | Train | Test | Total | Train | Test | Total |
| Patients number | 59 | 40 | 99 | 87 | 58 | 145 |
| Age(years) | | | | | | |
| <40 | 1 | 1 | 2 | 4 | 3 | 7 |
| 41-50 | 3 | 4 | 7 | 10 | 10 | 20 |
| 51-60 | 14 | 7 | 21 | 26 | 13 | 39 |
| 61-70 | 12 | 12 | 24 | 27 | 19 | 46 |
| $\geq 70$ | 29 | 16 | 45 | 20 | 13 | 33 |
| Gender | | | | | | |
| Male | 38 | 26 | 64 | 73 | 48 | 121 |
| Female | 21 | 14 | 35 | 14 | 10 | 24 |
| BCLC stage | | | | | | |
| A | 6 | 5 | 11 | 46 | 30 | 76 |
| B | 14 | 9 | 23 | 33 | 22 | 55 |
| C | 39 | 26 | 65 | 8 | 6 | 14 |
| Tumor Size(cm) | | | | | | |
| 0-3 | 6 | 5 | 11 | 22 | 15 | 37 |
| 3-5 | 9 | 6 | 15 | 27 | 24 | 51 |
| >5 | 17 | 7 | 24 | 38 | 19 | 57 |
| NaN | 27 | 22 | 49 | 0 | 0 | 0 |

TABLE I
DISTRIBUTION OF THE ***TCIA-TACE-SEG*** AND ***OP*** DATASETS

## C. Implementation Details

We implemented all the models using PyTorch and trained them on an NVIDIA A100 GPU. Given the small size of the dataset, we split the data into a 60:40 ratio for training and testing. We performed 100 bootstrapping iterations on the test set to ensure robust evaluation. This helped estimate the performance metrics with higher confidence and reduce the variance in the results. We used a learning rate of 1e-4 and a batch size of 4 for training. We also employed the Stochastic Gradient Descent (SGD) optimizer to iteratively update the model parameters. Since CNN-based models tend to perform stably with SGD, and ReViT shares similar characteristics, we adopted SGD as our primary optimizer. For the ViT model, we selected a cube size of (32, 32, 32) to process the input data, allowing the model to generate $N = 6 \times 6 \times 6$ instances for subsequent analysis. Additionally, recognizing the imbalance in class distribution, we utilized a balanced sampler to ensure that each class contributed equally during training.

## D. Baselines

We validated various models based on different design paradigms to establish a comparative foundation for evaluating our proposed method.

**ResNet10** [3] is a convolutional neural network that employs residual connections to facilitate the training of deep networks by mitigating the vanishing gradient problem. Additionally, we also utilized **AD3D-MIL** [26], a model that incorporates the principles of MIL specifically tailored for 3D medical imaging tasks. It enhances the capability to handle 3D CT volumes by treating each volume as a bag of instances and focusing on the most informative regions.

**Swin-Transformer** [20] employs shifted windows for self-attention, allowing the model to efficiently capture local and global features. This approach reduces computational complexity while retaining the benefits of transformers and mimicking the properties of CNNs. We utilized the encoder from Swin-UNETR and initialized the Swin Transformer in our setup using the pretrained weights from Swin-UNETR.

**MaxViT** [21] combines the strengths of CNNs and transformers through a multi-scale, multi-stage design. It captures both local and global spatial interactions using a four-stage

hierarchical structure with distinct self-attention mechanisms. This design enhances its ability to model long-range dependencies while maintaining computational efficiency. For the MaxViT model, we used the small variant without pre-trained weights.

### E. Performance Metrics

We evaluated all baseline models using standard performance metrics, including Macro-F1, Recall, Precision, and Accuracy. These metrics were chosen to provide a comprehensive assessment of each model's ability to accurately diagnose BCLC staging and to compare the performance across different approaches. Macro-F1 is particularly important in our evaluation because it provides a balanced measure of performance across all classes. We aim to ensure that our models are robust and reliable in accurately staging BCLC for all patients, thereby supporting more equitable and effective clinical decision-making.

## V. RESULTS & DISCUSSIONS

### A. Performance Comparison of Different Preprocessing Methods and Models

In this study, we comprehensively evaluated various preprocessing methods and models for BCLC staging using 3D abdominal CT images. The results, summarized in Table II, compare the Resized Masking and MCP methods across different models, including ResNet10, Swin-Transformer, and a hybrid ReViT model.

The Resized Masking method provided a straightforward approach to standardizing input sizes. However, despite its simplicity, this method resulted in relatively suboptimal performance. The primary reason for this is likely the significant loss of information and distortion caused by resizing the CT volumes from their original dimensions to a smaller, fixed size. This loss of crucial anatomical details adversely affects the model's ability to make accurate predictions, highlighting the limitations of this preprocessing technique.

In contrast, the MCP method demonstrated markedly superior performance. Inspired by the MIL framework, this approach involves preserving the original liver features by employing symmetric padding and central cropping to standardize the input size. By maintaining the anatomical integrity of the liver and focusing on the ROI, this method effectively mitigates the issues of information loss and distortion. Both the ResNet10 and Swin-Transformer models showed an overall improvement in performance with this preprocessing technique, indicating its effectiveness in enhancing model capabilities.

Our proposed hybrid ReViT model exhibited the best performance among all the tested models and preprocessing methods. This can be attributed to several key observations. First, we noticed that the AD3D-MIL model, which introduces the concept of MIL, performed better than the purely CNN-based ResNet10. This improvement underscores the value of MIL by focusing on the most informative instances. However, the BCLC staging task critically depends on accurately assessing the size and number of tumors in specific regions,

a requirement that places a high premium on detailed local feature extraction.

In this context, Transformer-based models, with their self-attention mechanisms, provide superior global context information, which is crucial for comprehensive image analysis. Among the Transformer-based models, the Swin-Transformer, which employs a shifted window self-attention mechanism, effectively mimics the properties of CNNs, enabling it to excel in capturing both local and global features. The superior performance of the Swin-Transformer over MaxViT highlights the importance of strong local feature learning capabilities for generating accurate cube instance embeddings in BCLC staging tasks.

Given these findings, our ReViT model, which combines the local feature extraction strengths of CNNs with the global context modeling abilities of Vision Transformers, stands out. ReViT's ability to integrate these complementary strengths results in a more accurate and robust analysis, thereby achieving the best performance among the models evaluated in our study.

### B. Impact of Pooling Strategies on Model Performance

In Table III, we compared the performance of ResNet10, ViT, and our proposed ReViT model under different MIL pooling strategies. Specifically, we examined Global Average Pooling and TopK Pooling ($K = 25\%$) to understand their impact on model performance.

Aggregating information from fewer instances is anticipated to result in a decline in model performance. Our findings support this expectation, as the TopK Pooling strategy generally leads to a performance drop. However, if the model can maintain performance using only the top-k most important instances, it indicates a strong capability to learn effective cube representations. This proficiency in filtering out less informative instances while preserving overall performance enhances both the accuracy and interpretability of the model. The motivation for using TopK Pooling lies in its potential to focus on the most relevant features, and we will further discuss the trade-off between performance and explainability in subsequent sections.

The experimental results show that the Macro-F1 score of ResNet10 decreased by 12% after applying TopK Pooling, while ViT experienced a 13% drop. In contrast, our proposed the ReViT model was able to maintain comparable performance even after discarding most instances during the aggregation process. These findings indicate that ResNet10's feature extraction without considering the global spatial interactions between cubes negatively impacts the BCLC staging task. On the other hand, ViT's use of only a single 3D convolution layer for linear projection limits its ability to learn local features effectively, resulting in suboptimal cube instance representations.

Our hybrid ReViT model, however, successfully addresses these limitations by combining the local feature extraction strengths of CNNs with the global context modeling capabilities of Vision Transformers. This integration allows ReViT to create more robust and informative cube instance representations, thereby maintaining high performance even with the

| Preprocessing Methods | Models | Params | Macro-F1 | Recall | Precision | Accuracy |
|---|---|---|---|---|---|---|
| Resized Masking [10] | ResNet10 | 14.36 M | 0.534 ± 0.050 | 0.535 ± 0.057 | 0.540 ± 0.048 | 0.541 ± 0.063 |
| | Swin-Transformer | 18.85 M | 0.565 ± 0.048 | 0.583 ± 0.073 | 0.580 ± 0.032 | 0.592 ± 0.068 |
| MCP | ResNet10 | 14.36 M | 0.538 ± 0.089 | 0.567 ± 0.080 | 0.556 ± 0.111 | 0.575 ± 0.091 |
| | Swin-Transformer | 18.85 M | 0.580 ± 0.103 | 0.594 ± 0.095 | 0.595 ± 0.118 | 0.604 ± 0.097 |
| | AD3D-MIL [26] | 0.32 M | 0.553 ± 0.103 | 0.573 ± 0.102 | 0.557 ± 0.113 | 0.582 ± 0.106 |
| | MaxViT [21] | 68.49 M | 0.561 ± 0.086 | 0.587 ± 0.076 | 0.573 ± 0.098 | 0.595 ± 0.091 |
| | ReViT (Ours) | 64.82 M | **0.634 ± 0.103** | **0.646 ± 0.096** | **0.637 ± 0.109** | **0.655 ± 0.104** |

| Models | Pooling Methods | Macro-F1 | Recall | Precision | Accuracy | Decrease (%) Macro-F1 | Accuracy |
|---|---|---|---|---|---|---|---|
| ResNet10 | Avg. | 0.538 ± 0.089 | 0.567 ± 0.080 | 0.556 ± 0.111 | 0.575 ± 0.091 | - | - |
| | TopK. | 0.471 ± 0.078 | 0.560 ± 0.064 | 0.522 ± 0.261 | 0.576 ± 0.093 | -12.45 | 0.17 |
| ViT | Avg. | 0.593 ± 0.096 | 0.615 ± 0.088 | 0.604 ± 0.112 | 0.624 ± 0.100 | - | - |
| | TopK. | 0.514 ± 0.075 | 0.582 ± 0.065 | 0.572 ± 0.159 | 0.593 ± 0.087 | -13.3 | -5.0 |
| ReViT (Ours) | Avg. | 0.634 ± 0.103 | 0.646 ± 0.096 | 0.637 ± 0.109 | 0.655 ± 0.104 | - | - |
| | TopK. | 0.633 ± 0.100 | 0.640 ± 0.099 | 0.635 ± 0.102 | 0.642 ± 0.103 | **-0.16** | **-1.98** |

TopK Pooling strategy. Additionally, ReViT's design fits well within the MIL framework, effectively handling the BCLC staging task by focusing on the most informative instances.

### C. Impact of Pooling Strategies on Model Explainability

Despite ReViT's superior performance with Global Average Pooling, it fails to provide visualizations that align with clinical relevance (see Figure 4). Therefore, in this ablation study, we will explore the impact of varying TopK ratios on model performance, as shown in Table IV.

Our findings show that Global Average Pooling, despite yielding high-performance metrics, produces visualizations that lack clinical relevance. This happens because the model, when trained to consider a large number of instances simultaneously, dilutes the impact of key instances. Conversely, TopK Pooling, by limiting the number of instances considered, allows the model to concentrate on the most critical regions. This approach directs the model's attention specifically towards liver tumors rather than the entire liver region, resulting in visualizations that align more closely with clinical insights and physician expectations.

These results highlight the trade-off between model performance and interpretability when using the TopK Pooling strategy with ReViT. While reducing $K$ can lead to a drop in overall performance metrics, the ability of TopK Pooling to enhance the model's focus on critical tumor regions demonstrates a significant advantage in clinical applications. In particular, our experiments and the heatmaps show that a $K$ value of 25% strikes the best balance between maintaining model performance and improving interpretability. This suggests that TopK Pooling can improve the practical utility of the model by making its predictions more aligned with clinical needs, even at the expense of some performance loss.

### VI. CONCLUSION

In this study, we introduced ReViT, a hybrid model that combines the strengths of CNNs and ViTs for BCLC staging

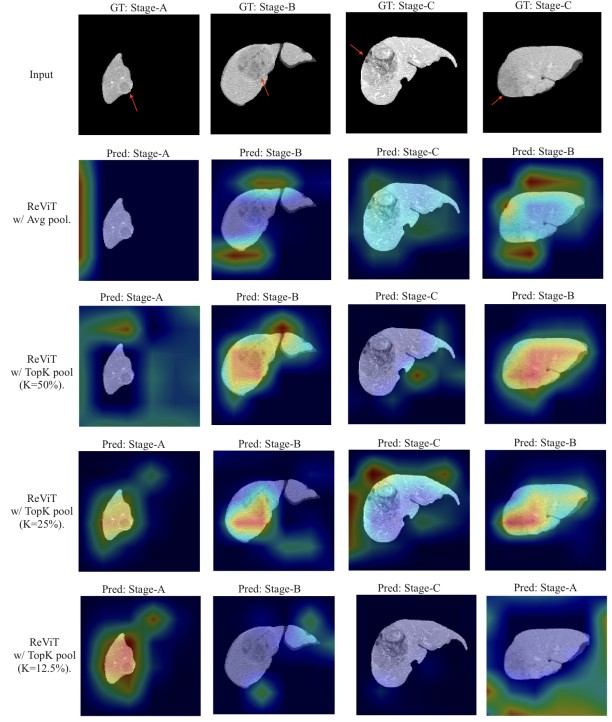

Fig. 4. Comparison of Grad-CAM heatmaps generated with different $K$ values. The visualizations demonstrate that when $K = 25\%$, the model's attention aligns most closely with clinically relevant regions as identified by physicians, focusing effectively on the liver tumors (see red arrows).

of HCC using 3D abdominal CT images. Leveraging the MIL framework, ReViT effectively handles the complexity and variability inherent in medical imaging tasks. The MCP technique preserves the liver's structural integrity, ensuring consistent input sizes. Coupled with the MIL framework and TopK Pooling strategy, this method enables ReViT to focus on the most informative liver regions, enhancing clinical relevance and providing clearer visualizations. Our results demonstrate that ReViT outperforms traditional CNN-based and Transformer-based models. The combination of local fea-

TABLE IV
IMPACT OF VARYING PROPORTIONS OF SELECTED INSTANCES ON PERFORMANCE

| Models | TopK | Macro-F1 | Recall | Precision | Accuracy | Decrease (%) | |
| --- | --- | --- | --- | --- | --- | --- | --- |
| | | | | | | Macro-F1 | Accuracy |
| ReViT (Ours) | K=100% | 0.634 ± 0.103 | 0.646 ± 0.096 | 0.637 ± 0.109 | 0.655 ± 0.104 | - | - |
| | K=50% | 0.601 ± 0.094 | 0.626 ± 0.083 | 0.622 ± 0.106 | 0.635 ± 0.098 | -5.21 | -3.05 |
| | K=25% | 0.633 ± 0.100 | 0.640 ± 0.010 | 0.635 ± 0.102 | 0.642 ± 0.103 | **-0.16** | **-1.98** |
| | K=12.5% | 0.609 ± 0.096 | 0.632 ± 0.092 | 0.624 ± 0.108 | 0.639 ± 0.097 | -3.94 | -2.44 |

ture extraction from CNNs and global context modeling from ViTs allows ReViT to maintain robust performance even with fewer instances, enhancing both accuracy and explainability. Future work will focus on expanding the dataset size to further improve model performance and generalizability, aiming to enhance clinical outcomes and diagnostic accuracy.

## ETHICS STATEMENT

The in-house dataset used in this study was approved by the NTU Hospital Ethics Committee under the IRB number 202306004RINC.

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
