# OpenReview forum: "ReViT: A Hybrid Approach for BCLC Staging of Hepatocellular Carcinoma Using 3D CT with Multiple Instance Learning"
_IEEE.org/EMBS/BHI/2024/Conference — IEEE BHI'24_

### Official Review · Reviewer_iYkr · 2024-08-12
**A Hybrid Approach for BCLC Staging of Hepatocellular Carcinoma  - Review**

**Overall Rating:** 6
**Confidence:** 5

**Other Quality Metrics:**

(a) Clarity of writing - Good
(b) Clinical Significance - Great
(c) Methodological Novelty - Good
(d) Experiments and Results - Good

**Questions For The Authors:**

1. Should we have a summary of detailed parameters for the models (total parameters, training times, inference times)? Based on that, we can see the trade between model complexity and efficacy.

2. Should we have a summary table of fine-tuned hyperparameters for the experiments conducted? This would help make the model more applicable in terms of reproducibility.

3. Minor comments: Each model was used; what variant was employed? For example, Swin Transformers has different variants, like based, small, medium, or large. Was the pre-trained weight used? How was it used?

**Strengths:**

The most important part of the work is proposing Multiple-Instance Learning that can flexibly work with CNN-based or ViT-based models.
The authors also show the end-to-end framework for demonstrating an automated BCLC staging system by applying the proposal.

**Summary Of The Paper:**

The idea and the methodology are novel.
It will contribute to the impact of employing the framework for clinical decision support in practice.

**Weaknesses:**

The details of the backbone models and the set of hyperparameters were lacking so that the audience has a better idea of how to reproduce the work in their application if applicable.

---

### Official Review · Reviewer_z9zf · 2024-08-12
**ReViT: A Hybrid Approach for BCLC Staging of Hepatocellular Carcinoma Using 3D CT with Multiple Instance Learning**

**Overall Rating:** 7
**Confidence:** 3

**Other Quality Metrics:**

(a) Clarity of writing: great
(b): Clinical significance: great
(c): Methodological novelty: great
(d): Experiments and results: good

**Questions For The Authors:**

- Why was ResNet10 selected as the pre-trained model for early feature extraction? Do any foundational models for medical image tasks exist, and if so, were these considered as an alternative?
- Despite the discussion, I still am having trouble understanding the explanation for why the Grad-CAM visualizations become more "interpretable" with decreasing K, despite decreasing model performance. Is this a case where perhaps Grad-CAM is not fully capturing the intricacies of your model's decision making process? I find it hard to believe that the highlighted regions in the heatmap for global average pooling are actually increasing the model's performance. Seems to me that either these heatmaps are misleading, or the minor advantage in the performance metrics are spurious.

**Strengths:**

- Hybrid approach based on a multiple instance learning framework
- Considering the 3-dimensional structure of the liver is an advantage for this task.
- Including the entire liver ROI rather than the tumor ROI makes sense to capture vascular invasion information in the model.

**Summary Of The Paper:**

This paper proposes a novel machine learning technique to stage HCC from abdominal CT scans according to the BCLC system, using a hybrid framework which combines multiple instance learning, vision transformers, and convolutional neural networks. Another novelty of this system is the use of 3-dimensional CT image data, whereas previous approaches have used independent 2-dimensional slices as training examples.

**Weaknesses:**

- I think this would be improved by a better understanding/explanation of the discrepancy between the Grad-CAM heatmaps (interpretability) and model performance.
- Results could be expanded, if there is space in the paper. For example, I am curious what the actual distribution of predicted versus actual classes on the test data looks like (which could be demonstrated in a heatmap). Were certain classes particularly difficult to classify, and if so, why might that be the case?

---

### Official Review · Reviewer_MFrh · 2024-08-12
**ReViT for Liver Cancer Staging**

**Overall Rating:** 7
**Confidence:** 4

**Other Quality Metrics:**

(a) Clarity of writing - Great
(b) Clinical Significance - Fair
(c) Methodological Novelty - Good
(d) Experiments and Results - Good

**Questions For The Authors:**

Why is SGD the optimizer of choice?

Would it help to normalize the orientation of the masked Liver ROI using PCA before applying padding in MCP?

A quantitative comparison of the proposed method with existing SotA for the same problem would help with better understanding performance of the proposed method in context of the liver cancer staging subdomain. The included metrics are comprehensive as far as the expectation of an ablation study would go but would those also be equivalent comparisons that can be considered replicating SotA methods on benchmark datasets?

**Strengths:**

Problem and method choices are well-motivated.
Results are promising. The use of Topk pooling despite the tradeoff resulting in more meaningful GradCAM is particularly interesting.
Most questions that come up while reading the manuscript are eventually answered.
Language is good.
Figures are good.
Understanding of SotA methods is satisfactory.

**Summary Of The Paper:**

Liver cancer staging is conventionally performed by experienced radiologists who inspect cross sections of volumetric CT abdominal scans and bucket cases by tumor count and size to indicate severity. Based on this staging, treatment plans are formulated and prognoses are made. Automating this process would result in significant speedup of the diagnostic-treatment loop but has been challenging owing to the scarcity of data. To solve this problem the authors frame the problem as a Multi Instance Learning task which allows the use of volume level labels to train models that can make predictions at more granular levels. This is done by first performing feature extraction using 3D CNN-based models to capture local spatial patterns and using the output feature volumes as inputs to a ViT based model that treats cubes of the feature volume tensor as tokens. This allows the modeling of global patterns within volumes and the dataset's distribution as a whole by embedding strong local features into a global space. The use of only those features that are correlated with what radiologists would look for is implicitly enforced by topk-pooling CNN-extracted features (k determined empirically). Furthermore the CLS token that is prepended to input tokens when training ViTs is omitted to prevent the model from considering every token from a volume as belonging to the volume-level class-label. The paper also addresses some inconsistencies commonly introduced in datasets during preprocessing with a masking and padding strategy that preserves variations in organ size.

**Weaknesses:**

A model that is sub 70% accuracy is at best a step towards automating liver cancer staging.

There seem to be a few too many parts that have to do with transformers being good at learning long-range dependencies.

The differences in workflow between MIL and regular classification could be made a bit clearer earlier in the paper. Specifically, what change in the problem formulation is it that makes the problem tractable? A one-to-one comparison of the step with the difference in formulation would help, in my opinion. This could be expressed concisely along the lines of:

> Given a dataset of 3D volumes representing tumors, each volume is labeled as one of three BCLC stages:
>         * Stage 0: Very Early Stage ($y_i = 0$)
>         * Stage A: Early Stage ($y_i = 1$)
>         * Stage C: Advanced Stage ($y_i = 2$)
>
>
> The objective is to learn a function that can classify sub-volumes (or "cubes" for short) of these tumors into one
> of the three BCLC stages.
>
>
> In regular multiclass classification, we assume that each individual data point (e.g., each sub-volume) has a single label associated with it. The goal is to learn a function $f(x)$ that maps input features $x$ to one of the $C=3$ classes. Mathematically, this is formulated as:
>
> $$p(y|x) = \frac{1}{Z} \cdot f(x)^y \cdot (1-f(x))^{C-y},$$
>
> where $f(x)$ is the sub-volume-level predictor, and $Z$ is a normalization constant.
>
> In contrast, Multiple Instance Learning (MIL) assumes that each individual data point (e.g., each tumor volume) contains multiple "bags" or instances, where each instance has an associated label. The goal in MIL is to learn a function $f(x)$ that maps input features $x$ to one of the classes, based on the collective behavior of all instances within the bag. Mathematically, this is formulated as:
>
> $$p(y|V) = \frac{1}{Z} \cdot \prod_{i=1}^{m} f(x_i)^y \cdot (1-f(x_i))^{C-y},$$
>
> where $f(x_i)$ is the instance-level predictor, and $Z$ is a normalization constant.
>
> The key difference between the two formulations: in MIL, the label associated with each data point is a function of the collective behavior of all instances within the bag, rather than just the individual instance itself.

---

### Decision · Program_Chairs · 2024-09-23

Accept